# Climate Change Influences the Population Density and Suitable Area of *Hippotiscus dorsalis* (Hemiptera: Pentatomidae) in China

**DOI:** 10.3390/insects14020135

**Published:** 2023-01-28

**Authors:** Mingzhen Zhao, Qian Duan, Xiayang Shen, Shaoyong Zhang

**Affiliations:** Key Laboratory of Vector Biology and Pathogen Control of Zhejiang Province, College of Life Science, Huzhou University, Huzhou 313000, China

**Keywords:** climate change, *Hippotiscus dorsalis*, damage investigation and analysis, potential distribution, distribution forecast

## Abstract

**Simple Summary:**

The damage by *Hippotiscus dorsalis* (Hemiptera: Pentatomidae) has resulted in the substantial mortality of moso bamboo in South China. In this study, we survey the damage of *H. dorsalis* from 2005 to 2013 in Huzhou, Zhejiang Province, and evaluate the current and future potential distribution in China. The results revealed (1) the April mean temperature and April maximum temperature were the main factors affecting the damage of *H. dorsalis*, and (2) the significant expansion of high suitable areas in Anhui and Jiangxi Provinces under future climate circumstances.

**Abstract:**

*Hippotiscus dorsalis* is the main pest of *Phyllostachys edulis* in South China. The relationship between climate change and outbreak of *H. dorsalis*, and the current and future distribution of *H. dorsalis* are unknown. This study aimed to confirm the effect of climate on population density and the attacked bamboo rate of *H. dorsalis*, using field survey data from 2005 to 2013 in Huzhou, Zhejiang Province, and to reveal the potential distribution of *H. dorsalis* under current and future climate conditions using the MaxEnt model. The damage investigation and distribution forecast revealed the following: (1) The mean monthly temperature and maximum temperatures were main factors affecting the population density and the attacked bamboo rate in April in the Anji county of Zhejiang Province; they are all significantly and positively correlated. (2) High suitable area will significantly expand in Anhui and Jiangxi Provinces under the future climate circumstances, and the total suitable area will present a decrease because of the precipitation restriction. The significant expansion of high suitable area in the Anhui and Jiangxi Provinces under future climate circumstances means that the affected provinces will face even greater challenges. These findings provide a theoretical basis for the early forecasting and monitoring of pest outbreaks.

## 1. Introduction

Moso bamboo (*Phyllostachys edulis*) is a critically economically important bamboo and is widely distributed in South China [1]. There are three kinds of pests that harm moso bamboo: bamboo shoots-eating (such as *Otidognathus davidis*), leaf-eating (such as *Ceracris kiangsu*), and stem-eating (such as *Hippotiscus dorsalis*) pests. *Hippotiscus dorsalis* (Hemiptera: Pentatomidae) is one of the main moso bamboo stem-eating pests, and it also can harm *Phyllostachys sulphurea*, *Phyllostachys glauca*, and so on [2].

The life history and external physical characteristics of *H. dorsalis* has already been described [3,4,5,6,7]. *H. dorsalis* has three developmental stages: egg, nymph, and adult. The nymph stage is divided into five instars. Its overwintering stages are mainly third-instar and fourth-instar nymph; the nymphs feed on bamboo from late March to early April. Adult emergence occurs during April to June; the second-generation eggs emerge in June, and the second-generation nymph overwintering begins in October. 

Climate change has had a profound impact on all aspects of terrestrial ecosystems and human beings [8,9]. Since the Industrial Revolution, an increase in global temperatures of 1 °C to 3.5 °C by the end of this century will lead to sea-level rises, the melting of Arctic ice, a range of changes in flora and fauna, and extreme climate events [10,11]. Climate change has increased the pest scope of activity, reduced overwintering mortality, and increased the number of generations produced per year [12,13,14]. There is growing evidence to show that climate change will exacerbate losses to agriculture and forestry caused by pests [15,16]. As a result of climate warming, there is a distribution shift towards the North and South Poles. The southern green stinkbug spread 85 km north in 45 years in Japan [17]; Monarch butterfly *Danaus plexippus* L. in the western hemisphere exhibited a northward shift, but the survival rates will face a new challenge if food, natural enemy and so on are not adequate [18]. 

Species distribution is an essential part of population dynamics [19]. Species distribution data originate from the public literature, field observation records, museums, and together with climate and topography, they reveal the rule of species distribution [20,21]. Species distribution models (SDMs) are constructed for the prediction of species distribution, and they have the advantage of analyzing species potential and future distribution [22,23]. SDMs include Maximum Entropy (MaxEnt), Bioclimate analysis and prediction system (BIOCLIM), Bayesian approach, Ecological Niche Factor Analysis (ENFA), Generalized linear model (GLM), and Boosted Regression Tree (BRT) [24,25,26]. SDMs are based on species distribution and environment data for the estimation of current and future potential distributions, and are widely used in the protection of wildlife, the invasive monitoring of alien species, and the effects of climate change on species distribution and abundance [27,28]. SDMs have become a vital research tool in ecology, biogeography, and evolutionary biology. In SDMs, the simulation accuracy of MaxEnt is higher than in other models, and it is well acknowledged for its short running time, simple operation, stable running results, and small sample size required [29,30,31,32]. MaxEnt had been applied to predict the suitability of many diseases and pests; for example, *Diaphorina citri* Kuwayama, *Xylella fastidiosa*, and *Lissorhoptrus oryzophilus* Kuschel, and the simulation results are in good agreement with the actual distribution of species [33,34,35].

Although *H. dorsalis* is a serious pest of moso bamboo, the research on *H. dorsalis* mainly focuses on biological characteristics and chemical control at present [6]; little is understood of the impact of climate change on its species population, and current and future potential distribution. This study analyzes the collected damage data of *H. dorsalis* and climate data, and studies the relationship between the distribution of *H. dorsalis* and climatic variables using the MaxEnt model. The purpose of this study was to reveal the dominant climatic factors affecting the damage and distribution of *H. dorsalis*, to analyze the potential distribution areas of *H. dorsalis* in China, and to provide an important reference and theoretical basis for formulating reasonable prevention and control measures. 

## 2. Materials and Methods

### 2.1. Damage Investigation of Hippotiscus dorsalis and Climate Data Collection from 2005 to 2013

This study was carried out in Anji county, Huzhou City, Zhejiang Province, China (30°23′–30°53′ N, 119°14′–119°53′ E) (Figure 1 and Appendix A). Anji county is the home of bamboo in China; it is the top producer for moso bamboo and commercial bamboo stock in China. At the same time, it is also the most serious area for moso bamboo pests. There is only one generation in a year for *H. dorsalis* in Anji county; the peak period of bamboo feeding by *H. dorsalis* was observed in April during the field observation. Data (population density of *H. dorsalis* and attacked rate of bamboo) were collected everyday over the whole of April from 2005 to 2013. Gaowuling, Yuhua, Shangshugai, Yao, Gangkou, Dalukou, and Shicen village of Anji county were the specific investigation sites. A total of 4400 moso bamboo plants were randomly selected for statistical recording every year from 2005 to 2013. The air temperature was also collected at 2:00, 8:00, 14:00, and 20:00 every day over the whole of April from 2005 to 2013. A U thermometer (Hebei Dongtai Instrument Co., China) was used to obtain maximum and minimum daily temperature. The related data were used to provide the basis for the mean April temperature, the SD of the mean April temperature, the maximum temperature in April, the minimum temperature in April, and the April mean diurnal temperature variation.

### 2.2. The Collection of Hippotiscus dorsalis Distribution Data

Species distribution data and environmental variables are preconditions of MaxEnt model prediction. The data of *H. dorsalis* distribution were obtained through the following ways: (1) reports of every Chinese provincial and prefecture-level city Forestry Bureau; (2) the field observations of teams. A total of 470 distribution sites were acquired through the methods above, and were proofread and screened using the buffer zone analytical approach via ArcMap software [36,37]. A total 97 distribution sites were finally determined, and the distance between these species’ distribution sites was greater than 10 km. Twenty-eight distribution sites were obtained from reports conducted by the Forestry Bureau, sixty-nine distribution sites were obtained from field observations (Appendix A). According to the MaxEnt software operation requirements, species name, longitude, and latitude were entered into Excel and saved in CSV format.

### 2.3. Environmental Variables

Nineteen environmental variables were downloaded from www.worldclim.org (accessed on 10 May 2022), and were used to conduct the MaxEnt model. The spatial resolution of these environmental variables are 5.0 arc minutes. Variable layers were changed into ASCII format using ArcMap 10.4.1. The National Map was obtained from the National Fundamental Geographic Information System (http://www.ngcc.cn (accessed on 10 May 2022)). Future climate databases were downloaded from Climate Change, Agriculture and Food Security program (http://ccafs-climate.org/ (accessed on 10 May 2022)) to forecast the future potential distribution of *H. dorsalis* under different climate scenarios.

The reports of the Intergovernmental Panel on Climate Change (IPCC, AR5) define Representative concentration pathways (RCPs) as a new scenario, it contains RCP2.6, RCP4.5, RCP6.0, and RCP8.5. They, respectively, represent the total radiative forcing of 2100 as 2.6 W/m², 4.5 W/m², 6 W/m², and 8.5 W/m² higher than 1750 [38]. RCP2.6, RCP4.5, and RCP8.5 were usually chosen as the basic scenarios in some of the research [39]. RCP8.5 was regarded as an unlikely high-risk future scenario and was not selected in this study. This study selected scenarios with low levels of greenhouse gas emissions (RCP2.6) and middle levels of greenhouse gas emissions (RCP4.5) to model the future distributions of *H. dorsalis* in the 2050s and 2070s.

In order to prevent over-fitting, the correlations of 19 environmental variables were analyzed by ArcGIS and SPSS [39,40]. Using 97 valid distribution data and 19 environmental data, the specific values of 19 environmental variables of 97 distributed data were extracted via ArcGIS; the output was in TXT format and was converted to CSV format. SPSS was used to conduct Pearson correlation analysis on 19 variables, and the correlation coefficient matrix between variables was calculated. The absolute value of the correlation coefficient of greater than 0.9 was defined as being highly correlated [39,40]. In the group of high correlation variables, the relatively small biological significance was removed, and the independent variables with great biological significance were selected. A total of 16 variables were screened for the establishment of the model (Table 1). The 16 selected environmental variables were used to calculate the percentage contribution (Table 2). Six non-contributing environment variables were deleted; 10 contributive environmental variables were chosen for using in the MaxEnt model.

### 2.4. Species Distribution Modeling

MaxEnt v.3.3.1 (http://www.cs.princeton.edu/wschapire/MaxEnt (accessed on 10 May 2022)) was used to predict the current potential distribution and future distribution in the 2050s and 2070s of *H. dorsalis*. Because of the occurrence of databases from different observers and collection methods in the process of occurrence records acquisition, a ten-percentile training presence logistic threshold was used to avoid sampling bias [41].

All environmental layers were converted to ASC format using the Arc Tools function in ArcGis, and the acquired ASC format layers and the distribution data were input to the MaxEnt model. A total of 75% distribution data were randomly chosen for model training, and the remaining 25% data were used for model validation [42]. The output layer was set to Logistic (logistic output improves model calibration, so that large differences in output values correspond better to large differences in suitability), and the rest of the parameters were selected as the model defaults [43].

Area under the curve (AUC) of receiver operating characteristic (ROC) was widely applied to evaluate the predictive performance of different models [44,45,46]. The AUC value is the index of sensitivity and specificity calculated at different thresholds; its value range is [0, 1]. The closer the AUC value is to 1, the better the model performance is.

The default parameter settings of MaxEnt often result in a considerable difference concerning actual species distribution; parameter optimization is crucial for the prediction accuracy and reliability of the results [47]. The ENMeval tool (https://github.com/marlonecobos/kuenm (accessed on 5 October 2022); https://www.r-project.org/ (accessed on 5 October 2022) was used to obtain the parameter optimization [48]. Akaike information criterion correction was used to evaluate the fitting degree and complexity of different parameters [49]. The fitting degree of different parameters was evaluated by the difference between training and testing AUCs, 10% training omission rate, and minimum training presence omission rate. The most suitable parameter combination was determined by the method mentioned above.

This threshold was used to define the potential suitable habitat areas for *H. dorsalis*, and so suitable areas were divided into four grades: unsuitable habitat (0–0.07), low suitability habitat (0.07–0.25), moderate suitability habitat (0.25–0.5), and high suitability habitat (0.5–1) [41,50,51]. By multiplying the number of existing grid units with different suitability levels in the distribution map using spatial resolution, the areas and percentage of different suitability areas of *H. dorsalis* under current and future climate scenarios were calculated.

## 3. Results

### 3.1. Correlation Analysis between the Damage Index of H. dorsalis and the Climate Index of the Study Site

The population density, attacked bamboo rate, highest population density in single day, and highest attacked bamboo rate in a single day April from 2005 to 2013 were statistically analyzed (Figure 2). The population density of *H. dorsalis* fluctuated around 0.5 head/stem from 2005 to 2011, increased significantly after 2011, and reached its maximum in 2013. The attacked bamboo rate was kept below 20% from 2005 to 2010, and after that, the attacked bamboo rate increased continually and reached its highest in 2013. The highest population density in a single day reached a peak in 2007, and the numerical value increased continually from 2010 to 2013. The line chart tendency of highest attacked bamboo rate in a single day was similar to the highest population density in a single day, in the range of 2005 to 2012. The highest attacked bamboo rate in a single day was also reached a peak in 2007, and the difference was that the numerical value of 2013 was lower than that in 2012.

The mean April temperature (°C), the SD of the mean April temperature (°C), the maximum temperature in April (°C), the minimum temperature in April (°C), and the April mean diurnal temperature variation (°C) from 2005 to 2013 were analyzed statistically (Figure 3). The mean April temperature reached its lowest point in 2010, and came to a head in 2012 and 2013. The SD of the mean April temperature fluctuated from 2.58 to 4.21 from 2005 to 2013. The maximum temperature in April reached the lowest point in 2010 and reached the highest point in 2013. The minimum temperature in April changed in a fluctuating way, reached the lowest point in 2010, and came to a peak in 2008 and 2012. The April mean diurnal temperature variation fluctuated from 6.01 to 9.21, and it reached the highest and lowest points in 2005 and 2008, respectively. 

Based on the collection damage index of *H. dorsalis* and the climate index of the study site, Pearson correlation analysis was carried out via SPSS (Table 3). The correlation between the damage index of *H. dorsalis* and the climate index of the study site was as follows: (1) population density was significantly and positively correlated with mean April temperature and maximum temperature in April, respectively; (2) attacked bamboo rate was significantly and positively correlated with mean April temperature and maximum temperature in April, respectively. Other correlations in the damage index of *H. dorsalis* or in the climate index of the study site were as follows: (1) population density and attacked bamboo rate showed an extremely significant and positive correlation; (2) the mean April temperature and maximum April temperature were extremely significantly and positively correlated.

### 3.2. Occurrence Records of H. dorsalis

The occurrence records of *H. dorsalis* were mainly located in low altitude areas, included Zhejiang, Anhui, Jiangxi, Hunan, and Guizhou Provinces (Figure 4). Zhejiang Province, the site of *H. dorsalis* damage investigation and climate data collection with the most occurrence records was found, and it accounted for 67 percent of the total effective occurrence records. The occurrence records of *H. dorsalis* revealed that most of the reported sites of *H. dorsalis* were located in the northwest of Zhejiang Province, southeast of Anhui Province, and northeast of Jiangxi Province. This area is an extensive planting area for moso bamboo that is located in the middle and lower sections of the Changjiang River Plain. There is a sporadic distribution in Jiangxi, Hunan, and Guizhou Provinces.

### 3.3. Performance and Variables Selection of the MaxEnt Model

The AUC value of the test on the applicability of the MaxEnt model was 0.98, and the model prediction was excellent (Figure 5). Using Pearson correlation analysis of the environmental variables: bio5 (maximum temperature of the warmest month) and bio10 (mean temp of the warmest quarter) had an extremely significant positive correlation; bio6 (minimum temperature of the coldest month) and bio11 (mean temperature of the coldest quarter) had an extremely significant positive correlation; bio13 (precipitation of the wettest month) with bio12 (annual precipitation) and bio16 (precipitation of the wettest quarter) had an extremely significant positive correlation respectively. At last, bio5, bio6, and bio13 were eliminated from the 19 environment variables to prevent over-fitting (Table 1). At the same time, with consideration of the percentage contribution of the 16 selected environment variables in the MaxEnt model, the following 10 environment variables were chosen to ultimately determine the habitat suitability of *H. dorsalis*: bio18 (precipitation of the warmest quarter, 42% contribution), bio4 (temperature seasonality, 23% contribution), bio14 (precipitation of the driest month, 17.6% contribution), bio15 (precipitation seasonality, 8.1% contribution), bio1 (annual mean temp, 3.9% contribution), bio19 (precipitation of the coldest quarter, 3.3% contribution), bio2 (mean diurnal range, 1% contribution), bio3 (isothermality, 0.5% contribution), bio10 (mean temp of the warmest quarter, 0.4% contribution), and bio11 (mean temp of the coldest quarter, 0.2% contribution) (Table 2).

### 3.4. Current Potential Distribution of H. dorsalis

The current potential distributions of *H. dorsalis* were mainly located in southern Jiangsu, Zhejiang, northern Fujian, southern Anhui, Hunan, Jiangxi and southern Hubei Province, amounting to 60 × 10^4^ km^2^ (Figure 6). High, moderate, and low suitability areas occupying the land area of China were 11.81 × 10^4^ km^2^, 26.02 × 10^4^ km^2^, and 22.18 × 10^4^ km^2^, respectively. The current potential distribution of *H. dorsalis* lies to the south of the Yangtze River, and is in the distribution area of moso bamboo (Figure 1). The high suitable area was mainly located in Zhejiang and Hunan province; the moderate suitable area was mainly located in Hunan, Jiangxi, southern Hubei, southern Anhui, and southern Jiangsu; and the low suitable areas extended around the high and moderate suitable areas.

### 3.5. Future Potential Distribution of H. dorsalis

The future potential distribution area of *H. dorsalis* has significantly changed from the current potential distribution area (Figure 7 and Figure 8). Under RCP2.6 in the 2050s, the high suitable areas are located in the Hunan, western Jiangxi, Zhejiang, and southern Anhui Provinces (Figure 7A). Low, moderate, high, and total suitable areas occupying the land of China are 23.71 × 10^4^ km^2^, 19.01 × 10^4^ km^2^, 12.19 × 10^4^ km^2^, and 54.91 × 10^4^ km^2^, respectively; under RCP2.6 in the 2070s, the high suitable areas are located in the Hunan, western Jiangxi, Zhejiang, and southern Anhui Provinces (Figure 7B). Low, moderate, high, and total suitability areas occupying the land area of China are 23.71 × 10^4^ km^2^, 19.2 × 10^4^ km^2^, 10.46 × 10^4^ km^2^, and 53.37 × 10^4^ km^2^, respectively, and under RCP4.5 in the 2050s, high suitable areas are located in the Zhejiang and southern Anhui Provinces (Figure 7C). Low, moderate, high, and total suitability areas occupying the land area of China are 26.5 × 10^4^ km^2^, 14.59 × 10^4^ km^2^, 4.22 × 10^4^ km^2^, and 45.31 × 10^4^ km^2^, respectively; under RCP4.5 in the 2070s, high suitable areas are located in the Zhejiang and southern Anhui Provinces (Figure 7D). Low, moderate, high, and total suitable areas occupying the land area of China are 25.63 × 10^4^ km^2^, 13.15 × 10^4^ km^2^, 4.9 × 10^4^ km^2^, and 43.68 × 10^4^ km^2^, respectively. Under RCP 2.6, the moderate and total suitable areas decreased from the present day to 2050 and 2070; the low and high suitable areas increased from the present day to 2050 and 2070 (Figure 8A). Under RCP 4.5, the moderate, high, and total suitable areas decreased from the present day to 2050 and 2070; the low suitable areas increased from the present day to 2050 and 2070 (Figure 8B).

## 4. Discussion

In Southeast China, moso bamboo is the most widely grown species (Figure 1), and is well known for its high economic value. The control and management of moso bamboo pests restricting the healthy development of commercial moso bamboo forests is an urgent problem for yielding the maximum economic benefits of moso bamboo [52]. Previous research on *H. dorsalis* has mainly focused on biological characteristics and chemical control, and the relationship between damage patterns and climate change. Until the current study, the current potential distribution and the future potential distribution of *H. dorsalis* has been infrequently reported. This research revealed the distribution of *H. dorsalis* in China and its infection development dynamics under future climate change. It is helpful for comprehensive prevention and control that is carried out nationwide under a future climate model.

Based on the damage results of *H. dorsalis* (Figure 2), there is an extremely significant positive correlation between the population density and attacked bamboo rate (Table 3) and the climate data results (Figure 3). The April mean temperature and the April maximum temperature are the main factors affecting population density and the attacked bamboo rate. The main reason may be that April is the time when the *H. dorsalis* develops from 4th-instar nymph to adult. The higher mean monthly temperature and the maximum temperature, the faster *H. dorsalis* develop. At the same time, this will result in the damage by *H. dorsalis* increasingly greater. These research results are consistent with the research in *Halyomorpha halys* (Hemiptera: Pentatomidae), climate change directly can influence the population abundance and will add further complications for management of population growth [53]. Rising global temperatures will increase the damage done by stinkbugs [54]. Moreover, the values for the highest population density and the highest bamboo attack in 2007 were high. The more concentrating development of *H. dorsalis* may be relevant to precipitation and temperature.

The damage index was not correlated with the highest population density in a single day and highest attacked bamboo rate in a single day (Table 3). None of the April climatic variables explained these highest population densities and attacked bamboo rate. This could result from the anomalously high values of population density in a single day and highest attacked bamboo rate in a single day recorded in 2007 (Figure 2C,D), which have not been explained. Further research on drivers that affected these maxima may be required. Based on occurrence records of *H. dorsalis*, all of these recording sites were located at a latitude between 25 and 32 degrees north (Figure 4). However, a lot of bamboo cultivation also occurred in the northern provinces of latitude 32 degrees north and in the southern provinces of latitude 25 degrees north. Additionally, there is no related infected report for *H. dorsalis*. The reason may be suitable climatic conditions (such as precipitation of warmest quarter and driest month (mm) and temperature seasonality) of the middle and lower Changjiang River Plain provided the necessary conditions for the survival of *H. dorsalis*.

The accuracy and performance of different SDMs had some differences in distribution forecast of the species [55,56]; the MaxEnt model was chosen for the reason that it has advantages in the process of processing small sample data sizes [57]. From the contribution results of environment variables (Table 2), precipitation (bio18 (precipitation of the warmest quarter), bio14 (precipitation of the driest month), and bio15 (precipitation seasonality) is the primary focus for the outbreak of *H. dorsalis*. The MaxEnt model prediction of *Arma custos* (Hemiptera: Pentatomidae) showed that the annual mean temperature and annual precipitation are the major factors influencing the distribution, and show that areas with medium precipitation should be focused on in the future [58].

From the results of the current potential distribution suitability of *H. dorsalis* using the MaxEnt model (Figure 6), the high suitable area was mainly focused on Zhejiang and Hunan Province, and the distribution area was consistent with the occurrence records of *H. dorsalis*. Furthermore, the Jiangsu, Jiangxi, and Anhui Province had the high suitable area. 

There is a considerable amount of evidence that supports the notion that climate change can have a significant impact on the geographical distribution of species [14,59,60]. The strength of influence of climate change on species has a strong correlation with the location of the species [61]. Under all of the studied climate scenarios, the longitude distribution range of current and future potential distribution is basically consistent with the occurrence records of *H. dorsalis*. In other words, climate change has little effect on the range variation of the longitude distribution of *H. dorsalis*. The reasons for this situation may be as follows: (1) the moso bamboo is mainly distributed in the southeast region of China, and the area planted to the west gradually decreases; this is one of the reasons for the difficulty of westward expansion; (2) the west of the middle and lower Changjiang River Plain (the main planting area of moso bamboo) are the Qinling Mountains, Ba Mountain, and the Yunnan-Guizhou Plateau; the climate demonstrates a marked difference with the west of the middle and lower Changjiang River Plain. This may be another major reason for why *H. dorsalis* has been unable to expand westward. The latitude range of the current and future potential distribution is much wider, spreading north to Jiangsu, Jiangxi and Hubei Provinces. 

Furthermore, with regard to future potential distribution compared with current potential distribution, the high suitable areas are expansion from Zhejiang Province to Anhui Province and from Hunan Province to Jiangxi Province. Possible causes for this are as follows: (1) The north is the North China Plain; the terrain is flat and mostly planted with bamboo, and it provides geographical advantages for the expansion of *H. dorsalis*; (2) Climate change makes the northern climate warmer, providing climatic conditions for the survival of *H. dorsalis*. Under future climatic scenarios, the suitable area of *A. custos* moved towards the north in China [58]. The northern limit of the distribution range of the southern green stink bug, *Nezara viridula* (L.) (Heteroptera: Pentatomidae) in central Japan has shifted northwards by 85 km along with climate change [8]. These findings are consistent with the global potential range of *H. halys*, which will expand polewards under future climate scenarios [53]. Based on these research results, it is revealed that the activity range of shieldbugs gradually expands to a circumpolar latitude due to climate change.

In comparison to current potential distribution of *H. dorsalis*, the total suitable area will be slightly lower by 8.3% and 11.2% in 2050 and 2070 under RCP 2.6, attributed to the decline of moderate suitable area. The high and low suitable areas changed slightly (Figure 7 and Figure 8A). In comparison to the potential distribution of *H. dorsalis* at present, the total suitable area decreased by 24.3% and 27.2% in 2050 and 2070, respectively, under RCP 4.5, attributed to the decline in the moderate and high suitable area. The moderate suitable area decreased by 43.9% and 49.4% in 2050 and 2070, respectively, under RCP 4.5; the high suitable area decreased by 64.2% and 58.5% in 2050 and 2070 under RCP 4.5; and the low suitable area increased by 19.5% and 15.6% in 2050 and 2070, respectively, under RCP 4.5 (Figure 7 and Figure 8B). From the above comparison results, it is revealed that the total suitable area constantly decreases due to future climate circumstances. By combining the data of *H. dorsalis* damage in Anji county, it can be observed that climate change plays a key role in population density. At the same time, the temperature in the southern region led to unsuitable conditions for the survival of *H. dorsalis*; the overall-damage area showed a northward moving trend. Combining the contribution of 19 environment variables, precipitation contributed with 71% (Table 2 and Table 3); simultaneously, combining the potential distribution at present shows that the distribution range of *H. dorsalis* is located in the south of the Yangtze River. The poor precipitation levels in the north of the Yangtze River limited the northward movement of the damaged zone. The high-temperature limit in the southern damaged zone and precipitation restriction in the northern damaged zone may have been the main reasons for the decrease in the total suitable area.

Low population densities can occur in areas with low- or high-suitability values, whereas high population densities are restricted to areas where the suitability is better [62,63]. The significant increase in the high suitable area in the Anhui and Jiangxi Provinces created more geographic areas for the high population densities of *H. dorsalis*. This increased the probability of a *H. dorsalis* outbreak. This is a huge challenge for the improved prevention and control capacity of *H. dorsalis*. Additionally, the increase from low to high population may cause situations where areas are unprepared for the management and control of *H. dorsalis*. So, local forestry bureaus should include *H. dorsalis* in their regular investigations to prevent sudden outbreaks, and large-scale bamboo planting areas should take more precautionary measures. 

The core distributional area showed a significant shift tendency due to future climate change. The core distributional area shifts from Zhejiang to Anhui and from Hunan to Jiangxi, greater control and supervision of *H. dorsalis* needs to be implemented in these areas, to prevent serious damage to moso bamboo by large outbreaks of *H. dorsalis*. *H. dorsalis* feeds on the juices of the moso bamboo joint, and this is easy to observe from April to September on the moso bamboo joint. Additionally, *H. dorsalis* has been called “smelly fart pest” by bamboo farmers. During a *H. dorsalis* outbreak, the smell can be recognized from miles away.

Insect development depends on its host [64], so that it is of scientific significance to consider the host distribution range to predict the distribution of pests. Moso bamboo is the main host of *H. dorsalis* in China. Moso bamboo, as an important economic bamboo species in China, has brought great economic benefits and has protected the ecosystem [65]. The cultivated area of moso bamboo is to expand year after year [66]. This aspect will also aggravate the harm done by bamboo pests. There are many uncertainties in the forecast, especially from human factors in changing the planting area of moso bamboo. In addition, the hosts of *H. dorsalis* also include *P. sulphurea* and *P. glauca*, and these host bamboos are widely distributed in South China. Therefore, it is also very important to control the spread of *H. dorsalis* for bamboo because climate change influences the population density and suitable areas of *H. dorsalis*. Of course, there are some limiting factors in this study: (1) although MaxEnt has advantages in processing small size data, the small size still may influence the accuracy of results; (2) although the ENMeval tool is used to optimize the default parameter settings of MaxEnt, the predicted results may vary by other SMD models.

## 5. Conclusions

Precipitation was the main factor influencing the division of the suitable area of *H. dorsalis*. High population densities are restricted to areas with better suitability [63]. These findings revealed that precipitation was the main factor influencing the population densities in different distribution areas of moso bamboo. Moreover, the mean monthly temperature and maximum temperature were the primary factors affecting population density and attacked bamboo rate of Anji county (the representative of high suitable area) in April. High population densities were the specific manifestations of an outbreak in field observation. Thus, these results indicated that precipitation, the mean monthly temperature and maximum temperature were the main factors affecting the outbreak of *H. dorsalis*. Current study is possible to localize the outbreak area of *H. dorsalis* within the high suitable areas. However, further research about the effects of temperature and humidity on the development of *H. dorsalis* is needed to determine whether there is an outbreak in the specific high suitable area. These findings are conductive to predicting the occurrence and outbreak of *H. dorsalis* under future climatic conditions, and providing the basis for the prevention and control strategy of that year. According to these results, it should be possible to mount precautionary measures against the further expansions and outbreaks of *H. dorsalis*.

## Figures and Tables

**Figure 1 insects-14-00135-f001:**
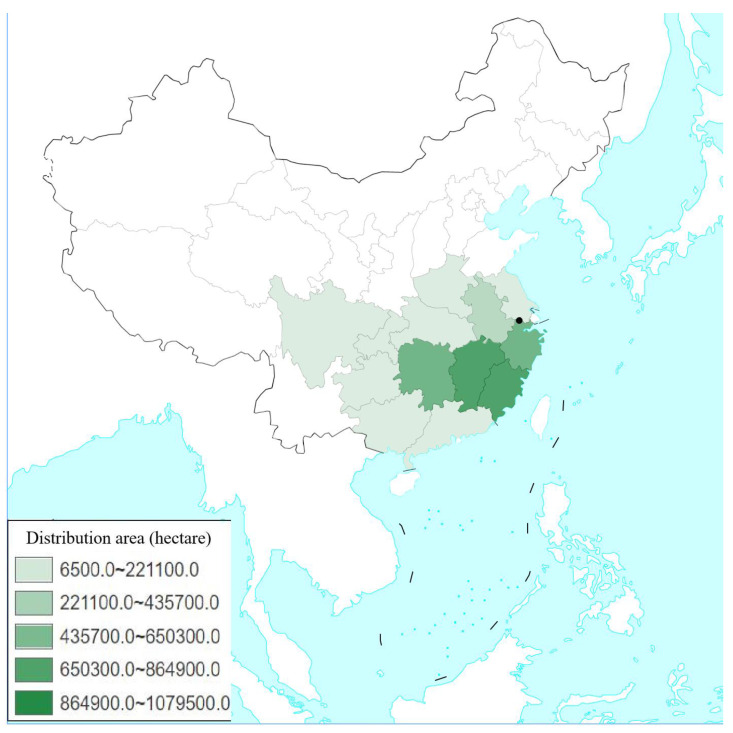
Distribution area (hectare) of *Phyllostachys heterocycla* in each province of China. The black spot represents the site of the hazard investigation of *Hippotiscus dorsalis* and climate data collection (Anji county, Huzhou City, Zhejiang Province, China). The data of moso bamboo distribution were based on the ninth national first class bamboo forest area, and the number of plants according to the forest species statistical table was downloaded from national forestry and grassland data center (http://www.forestdata.cn/index.html (accessed on 20 September 2020)).

**Figure 2 insects-14-00135-f002:**
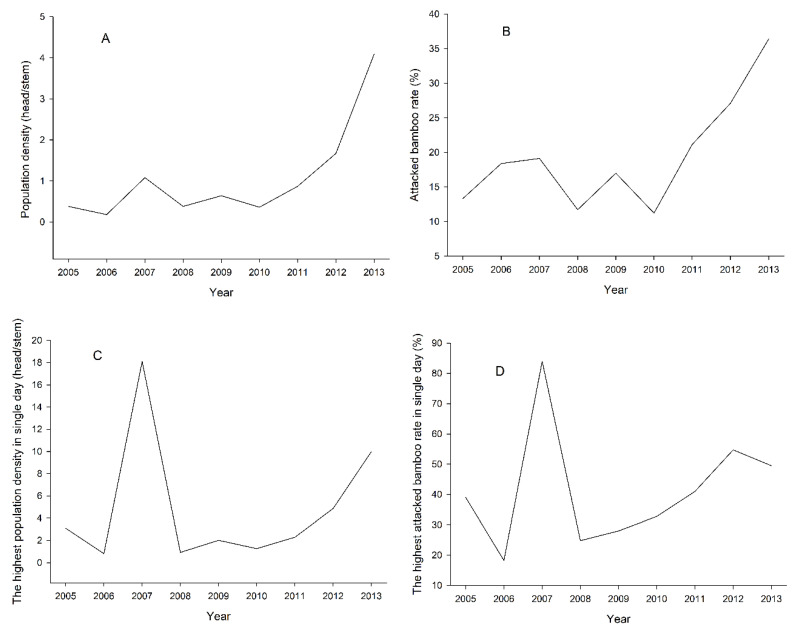
The population density (**A**), attacked bamboo rate (**B**), highest population density in single day (**C**), and highest attacked bamboo rate in single day (**D**) April from 2005 to 2013. The number of moso bamboo plants surveyed each year is 4400, and the survey site is located at Anji County, Huzhou City, Zhejiang Province, China.

**Figure 3 insects-14-00135-f003:**
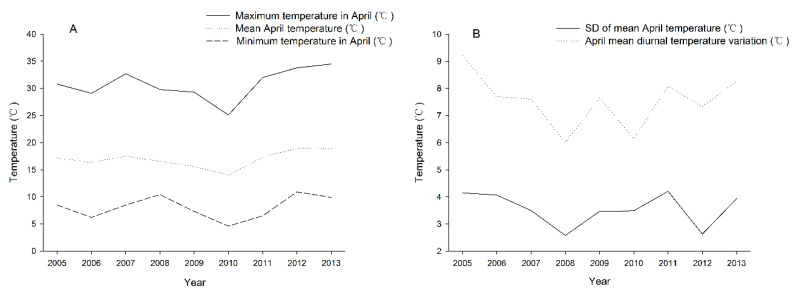
April climate data (2005–2013) collected from Anji County, Huzhou City, Zhejiang Province, China. (**A**): The mean April temperature (°C), the maximum temperature in April (°C) and the minimum temperature in April (°C). (**B**): the SD of the mean April temperature (°C) and the April mean diurnal temperature variation (°C).

**Figure 4 insects-14-00135-f004:**
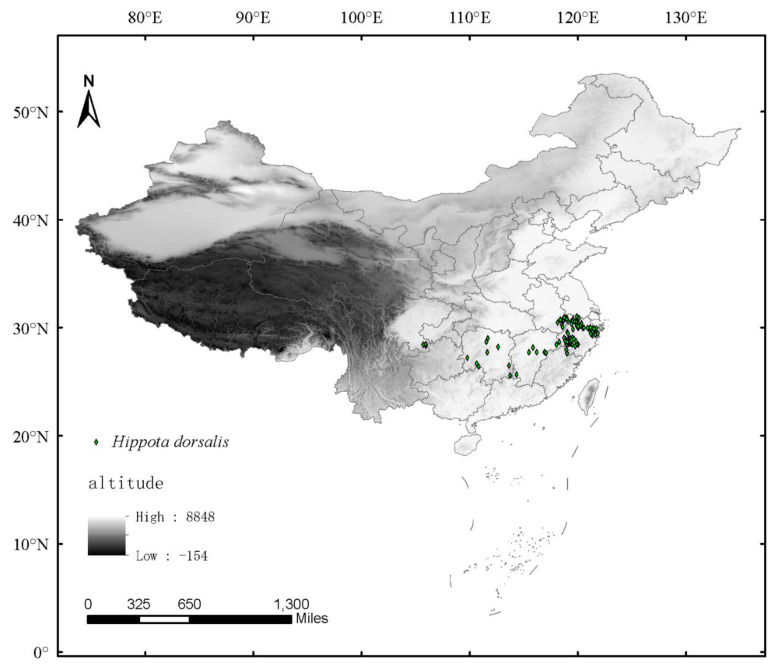
Occurrence records of *Hippotiscus dorsalis* in China.

**Figure 5 insects-14-00135-f005:**
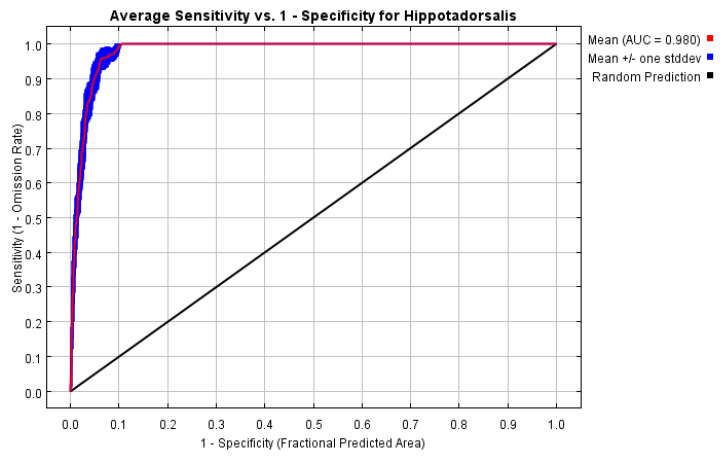
AUC value of test on applicability of the MaxEnt model.

**Figure 6 insects-14-00135-f006:**
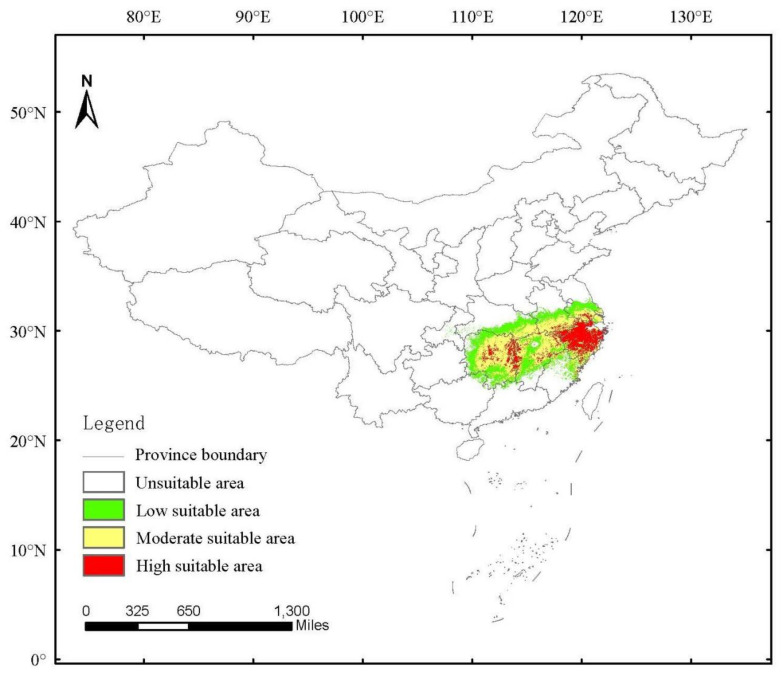
Present habitat distribution suitability of *Hippotiscus dorsalis*.

**Figure 7 insects-14-00135-f007:**
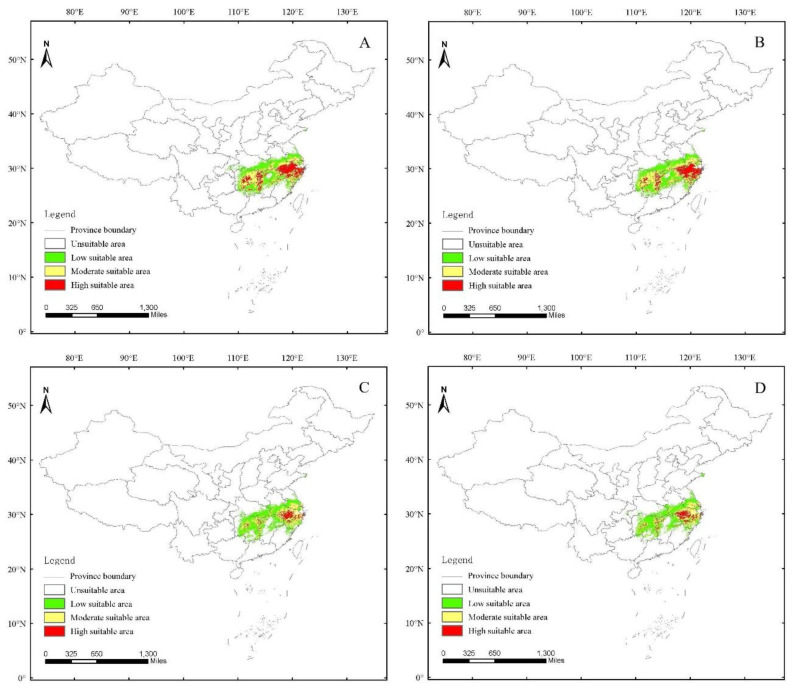
Future potential distribution of *Hippotiscus dorsalis* in China. (**A**) is future potential distribution of *Hippotiscus dorsalis* under RCP2.6 in 2050s; (**B**) is future potential distribution of *Hippotiscus dorsalis* under RCP2.6 in 2070s; (**C**) is future potential distribution of *Hippotiscus dorsalis* under RCP4.5 in 2050s; (**D**) is future potential distribution of *Hippotiscus dorsalis* under RCP4.5 in 2070s.

**Figure 8 insects-14-00135-f008:**
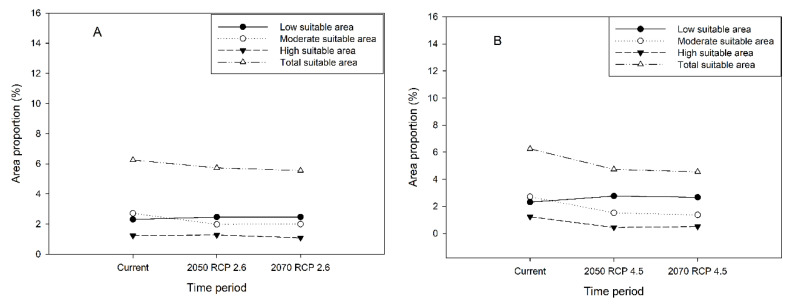
The change in area proportion (%) from current to 2070s under RCP2.6 (**A**) and under RCP4.5 (**B**).

**Table 1 insects-14-00135-t001:** Correlation analysis and screening of environmental variables.

Abbreviation	Environmental Variables	Pearson > 0.9	Operation
Bio1	Annual mean temp (°C)		Retain
Bio2	Mean diurnal range (°C)		Retain
Bio3	Isothermality		Retain
Bio4	Temperature seasonality		Retain
Bio5	Maximum temp of warmest month (°C)	Bio10	Eliminate
Bio6	Minimum temp of coldest month (°C)	Bio11	Eliminate
Bio7	Temperature annual range (°C)		Retain
Bio8	Mean temp of wettest quarter (°C)		Retain
Bio9	Mean temp of driest quarter (°C)		Retain
Bio10	Mean temp of warmest quarter (°C)	Bio5	Retain
Bio11	Mean temp of coldest quarter (°C)	Bio6	Retain
Bio12	Annual precipitation (mm)	Bio13	Retain
Bio13	Precipitation of wettest month (mm)	Bio16, Bio12	Eliminate
Bio14	Precipitation of driest month (mm)		Retain
Bio15	Precipitation seasonality (mm)		Retain
Bio16	Precipitation of wettest quarter (mm)	Bio13	Retain
Bio17	Precipitation of driest quarter (mm)		Retain
Bio18	Precipitation of warmest quarter (mm)		Retain
Bio19	Precipitation of coldest quarter (mm)		Retain

For example, as for lines 6 (Bio5) and 11 (Bio10), the Pearson correlation coefficients between Bio5 and Bio10 are greater than 0.9. The eliminating operation is performed for Bio5, and the retaining operation is performed for Bio10.

**Table 2 insects-14-00135-t002:** The percentage contribution and permutation importance of environmental variables for predicting the distribution of *Hippotiscus dorsalis*.

Variable	Percent Contribution	Permutation Importance
Bio18	42	1.5
Bio4	23	0
Bio14	17.6	0.7
Bio15	8.1	0
Bio1	3.9	0.6
Bio19	3.3	1.2
Bio2	1	0.4
Bio3	0.5	57.2
Bio10	0.4	1.6
Bio11	0.2	36
Bio17	0	0.4
Bio8	0	0.1
Bio9	0	0.4
Bio16	0	0
Bio12	0	0
Bio7	0	0

**Table 3 insects-14-00135-t003:** The Pearson correlation significance values between damage index of *Hippotiscus dorsalis* and climate index of study site.

	01	02	03	04	05	06	07	08	09
01	1								

02	0.923 **	1							
	0.000								
03	0.488	0.432	1						
	0.182	0.245							
04	0.406	0.389	0.923 **	1					
	0.278	0.301	0.000						
05	0.679 *	0.784 *	0.483	0.529	1				
	0.044	0.012	0.188	0.143					
06	0.060	0.118	0.026	−0.075	−0.060	1			
	0.877	0.763	0.946	0.848	0.878				
07	0.698 *	0.792 *	0.579	0.605	0.976 **	−0.005	1		
	0.037	0.011	0.103	0.084	0.000	0.990			
08	0.488	0.452	0.323	0.345	0.782 *	−0.544	0.739 *	1	
	0.182	0.222	0.396	0.363	0.013	0.130	0.023		
09	0.273	0.376	0.226	0.215	0.481	0.708 *	0.515	0.089	1
	0.477	0.318	0.559	0.578	0.190	0.033	0.156	0.819	

Variables 01–04 are damage index of *H. dorsalis*, respectively representing population density, attacked bamboo rate, highest population density in single day, and highest attacked bamboo rate in single day April from 2005 to 2013. Variables 05–09 are climate index of study site, respectively, representing mean April temperature, SD of mean April temperature, maximum temperature in April, minimum temperature in April, and April mean diurnal temperature variation. The asterisks indicate significant differences (* *p* ≤ 0.05, ** *p* ≤ 0.01).

## Data Availability

Data is contained within the article or Appendix A.

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
