# Peer review of "Climate Change Influences the Population Density and Suitable Area of *Hippotiscus dorsalis* (Hemiptera: Pentatomidae) in China"

_insects, 2023, doi:10.3390/insects14020135_

Round 1

Reviewer 1 Report

The manuscript reports the dominant climatic factors affecting the damage and distribution of Hippotiscus dorsalis, and the potential distribution areas of H. dorsalis in China under current and future climate conditions. The authors explain reasonable information of introductions, materials and methods, results and discussions.

Major

1.    References are not enough to support the introduction and discussion.

2.    The resolution of figures is not clear.

3.    The authors give more discussion related to the results of this study.

Minor

What is Dr. Xiayang Shen’s affiliation?

Line 11: “from 2005 to 2013” à Please give some information why the authors investigated the density of H. dorsalis and attacked rate on bamboo from 2005 to 2013. After 2013, the issue of climate change is expanded. There is no data after 2013.

Line 22: delete “these”.

Line 25: RCP is abbreviation. Please give the full explanation.

Line 35: Reference 1 is inappropriate. Please give correct reference.

Line 36: bamboo shoots, leaves and branch (stems) pests.

Line 47: bamboo shoots or leaves or branch. Please give detailed information.

Line 51: Please check the reference number 10 because it does not describe about “Adults and nymph H. dorsalis suck moso bamboo sap on the moso bamboo joint, and cause 50 the death of moso bamboo”.

Line 55: [14-15] à [14,15].

Line 56: by the end of the century à by the end of this century?

Line 56: 1 C to 3.5 C à 1 to 3.5.

Line 58: References 16 and 17 are inappropriate. Please give correct references.

Line 60: Please provide recent references.

Line 74: In species distribution models à In SDMs.

Line 74-76: Please check the reference number 33 because it does not relate to the contents of manuscript.

Line 91: Please give the information of latitude and longitude of Anji county.

Line 94-95: Please give the detailed information. population density à population density of H. dorsalis.

Line 95-96: Please check “Gaowuling, Yuhua, Shangshugai, Yao, Gangkou, Dalukou, and Shicen village”.

Line 94-102: The authors explain why they choose the April from 2005 to 2013.

Figure 1: The resolution of figure legends is not clear. What is the small box of right corner in the figure?

Line 109: Please give the information when the authors access the website.

Line114-116: Please give the information when the authors access the website.

Line 119-120: Please give the reference about ArcMap software.

Line 125 and 129: Please give the information when the authors access the website.

Line 136-138: Please give some information why the authors choose RCP2.6 and RCP4.5.

Line 156: Please give the information when the authors access the website.

Line 159: A 10th  percentile training à A ten-percentile training

Line 160: Please give original reference instead of “38”.

Line 165: Logistic format à logistic format.

Line 168-169: “low habitat suitability, moderate habitat suitability and high habitat suitability: à “low suitability habitat, moderate suitability habitat, and high suitability habitat”

Figure 2: The resolution of figure is not clear. Please give the explanation of “A”, “B”, “C”, and “D”, respectively.

Line 186: “showed significant growth” à increased significantly?

Figure 3: The resolution of figure is not clear. Please give the explanation of “A”, “B”, “C”, “D” and “E”, respectively.

Line 214-230: Please rewrite the sentences.

Figure 4: Please explain why the authors show the figure legend of altitude.

Line 240-247: Please rewrite the sentences.

Table 2: Temperature annual range à Temperature annual range ().

Table 3: Please remake the table 3.

Figure 6: If possible, please show colorful legend for readers. Please delete “current” inside the figure box.

Figure 7: If possible, please show colorful legend for readers.

Line 278-284: Please rewrite the sentences.

Line 286-287: Please check the sentence. There is no period.

Line 297: “economic bamboo” is redundant.

Line 300: Please delete comma.

Line 308-318: Repeated the contents of results.

Line 319-321: Please give correct references.

Line 323-342: Repeated the contents of results.

Line 343-347: Discussion is needed to connect the contents of results.

Line 351-356: Repeated the contents of results.

Line 357: A. custos à Arma custos.

Line 365-367: These sentences do not connect front and rear sentence.

Line 373: “the main planting area of H. dorsalis” à “the main planting area of bamboo”?

Line 382-392: Please show the logical relationship between this study and references.

Line 420-423: Please rewrite the sentences. For example, “Therefore, it is also very important to control the spread of H. dorsalis for these bamboos because climate change will influence the population density and the high suitable areas of H. dorsalis”.

Line 427: “reveal” à “revealed”.

Line 429: “judging” à “determining”

References: The authors follow the guideline of “Journal Insects”.

Author Response

The manuscript reports the dominant climatic factors affecting the damage and distribution of Hippotiscus dorsalis, and the potential distribution areas of H. dorsalis in China under current and future climate conditions. The authors explain reasonable information of introductions, materials and methods, results and discussions.

Major

  1. References are not enough to support the introduction and discussion.

Revised. References were added in introduction and discussion.

  1. The resolution of figures is not clear.

When an image is placed in the document, the resolution decreases. Therefore, we have uploaded some pictures individually as a separate attachment.

  1. The authors give more discussion related to the results of this study.

 Discussion part was fully revised. The suggestion of reviewer No.2: This study uses the maxent model with default parameter settings, which can lead to inaccurate results of the model. It is recommended to use ENMeval tool for parameter optimization. Kass, JM, Muscarella, R, Galante, PJ, et al. ENMeval 2.0: Redesigned for customizable and reproducible modeling of species’ niches and distributions. Methods Ecol Evol. 2021; 12: 1602– 1608. https://doi.org/10.1111/2041-210X.13628. So ENMeval tool was used to execute parameter optimization. Figure 5-8 were reproduced by these new results, all related description had been revised throughout the article.

Minor

What is Dr. Xiayang Shen’s affiliation?

Revised. Line 4.

Line 11: “from 2005 to 2013” à Please give some information why the authors investigated the density of H. dorsalis and attacked rate on bamboo from 2005 to 2013. After 2013, the issue of climate change is expanded. There is no data after 2013.

A field investigation consumes a considerable amount of manpower and resources; it was therefore suspended due to a lack of financial support in 2014 and, for various other reasons, has not been continued. Nevertheless, the nine years of data collection are extremely valuable for analyzing the damage density and attacked rate on bamboo caused by the bamboo pest Hippotiscus dorsalis.

Line 22: delete “these”.

Revised. Line 23.

Line 25: RCP is abbreviation. Please give the full explanation.

Revised. The original sentence was removed due to the change in results.

Line 35: Reference 1 is inappropriate. Please give correct reference.

Revised. Line 616.

Line 36: bamboo shoots, leaves and branch (stems) pests.

Revised. Line 42.

Line 47: bamboo shoots or leaves or branch. Please give detailed information.

Revised. Line 42.

Line 51: Please check the reference number 10 because it does not describe about “Adults and nymph H. dorsalis suck moso bamboo sap on the moso bamboo joint, and cause 50 the death of moso bamboo”.

Revised.

Line 55: [14-15] à [14,15].

Revised.

Line 56: by the end of the century à by the end of this century?

Revised.

Line 56: 1 ◦C to 3.5 ◦C à 1℃ to 3.5℃.

Revised.

Line 58: References 16 and 17 are inappropriate. Please give correct references.

Revised. Line 657.

Line 60: Please provide recent references.

Revised. Line 673.

Line 74: In species distribution models à In SDMs.

Revised.

Line 74-76: Please check the reference number 33 because it does not relate to the contents of manuscript.

Revised. Lines 706-717.

Line 91: Please give the information of latitude and longitude of Anji county.

Revised. Line 105.

Line 94-95: Please give the detailed information. population density à population density of H. dorsalis.

Revised.

Line 95-96: Please check “Gaowuling, Yuhua, Shangshugai, Yao, Gangkou, Dalukou, and Shicen village”.

Revised.

Line 94-102: The authors explain why they choose the April from 2005 to 2013.

Revised. There is only one generation in a year for H. dorsalis in Anji county; April was observed as the peak period of bamboo feeding during the field observation. Related descriptions have been added to the original article. Line 107.

Figure 1: The resolution of figure legends is not clear. What is the small box of right corner in the figure?

Revised. The figure was update. Line 121.

Line 109: Please give the information when the authors access the website.

Revised.

Line114-116: Please give the information when the authors access the website.

Revised.

Line 119-120: Please give the reference about ArcMap software.

Revised. Line 140.

Line 125 and 129: Please give the information when the authors access the website.

Revised.

Line 136-138: Please give some information why the authors choose RCP2.6 and RCP4.5.

Revised. Line 159.

Line 156: Please give the information when the authors access the website.

Revised.

Line 159: A 10th percentile training à A ten-percentile training

Revised.

Line 160: Please give original reference instead of “38”.

Revised. Line 186.

Line 165: Logistic format à logistic format.

Revised.

Line 168-169: “low habitat suitability, moderate habitat suitability and high habitat suitability: à “low suitability habitat, moderate suitability habitat, and high suitability habitat”

Revised.

Figure 2: The resolution of figure is not clear. Please give the explanation of “A”, “B”, “C”, and “D”, respectively.

Revised.

Line 186: “showed significant growth” à increased significantly?

Revised.

Figure 3: The resolution of figure is not clear. Please give the explanation of “A”, “B”, “C”, “D” and “E”, respectively.

Revised. Since an image has been added to the document, the resolution has reduced. High-resolution images will be uploaded as attachments.

Line 214-230: Please rewrite the sentences.

Revised. Lines 258-278.

Figure 4: Please explain why the authors show the figure legend of altitude.

This design mainly points out the altitude range of distribution of H. dorsalis in China. H. dorsalis is mainly distributed in low-altitude areas. To provide a basis for the subsequent explanation for the pest not expanding to western China. Related descriptions have been added to the original article. Lines 280-289.

Line 240-247: Please rewrite the sentences.

Revised. Lines 294-302.

Table 2: Temperature annual range à Temperature annual range (℃).

Revised.

Table 3: Please remake the table 3.

Revised.

Figure 6: If possible, please show colorful legend for readers. Please delete “current” inside the figure box.

Revised.

Figure 7: If possible, please show colorful legend for readers.

Revised.

Line 278-284: Please rewrite the sentences.

Revised. Line 358.

Line 286-287: Please check the sentence. There is no period.

Revised.

Line 297: “economic bamboo” is redundant.

Revised.

Line 300: Please delete comma.

Revised.

Line 308-318: Repeated the contents of results.

Revised. Lines 416-428.

Line 319-321: Please give correct references.

Revised. Line 431.

Line 323-342: Repeated the contents of results.

Revised. Line 433.

Line 343-347: Discussion is needed to connect the contents of results.

Revised. Lines 455-461.

Line 351-356: Repeated the contents of results.

Revised. Line 465.

Line 357: A. custos à Arma custos.

Revised.

Line 365-367: These sentences do not connect front and rear sentence.

Revised. Line 493.

Line 373: “the main planting area of H. dorsalis” à “the main planting area of bamboo”?

Revised.

Line 382-392: Please show the logical relationship between this study and references.

Revised. Lines 502-515.

Line 420-423: Please rewrite the sentences. For example, “Therefore, it is also very important to control the spread of H. dorsalis for these bamboos because climate change will influence the population density and the high suitable areas of H. dorsalis”.

Revised.

Line 427: “reveal” à “revealed”.

Revised.

Line 429: “judging” à “determining”

Revised.

References: The authors follow the guideline of “Journal Insects”.

Revised.

Reviewer 2 Report

The manuscript analyzes the population density and high suitable area ratio of Hippotiscus dorsalis (Hemiptera: Pentatomidae) in China. The work has the potential of being an interesting study about the possible threat posed by H. dorsalis both in the present and in the future. However, several aspects of the paper need to be revised such as distribution data selection, methods explanation, etc. Moreover, the work lacks an ecological perspective and it would need a more in-depth analysis of bibliography regarding several topics. These are the main general issues:

This study contains two independent parts of research content, and the author does not connect the two parts. How to incorporate the population density change into the species distribution modelCombine the two parts is the main innovation of this study. (Tôrres, N.M., De Marco, P., Júnior, Santos, T., Silveira, L., de Almeida Jácomo, A.T. and Diniz-Filho, J.A.F. (2012), Can species distribution modelling provide estimates of population densities? A case study with jaguars in the Neotropics. Diversity and Distributions, 18: 615-627. https://doi.org/10.1111/j.1472-4642.2012.00892.x; Oliver, T.H., Gillings, S., Girardello, M., Rapacciuolo, G., Brereton, T.M., Siriwardena, G.M., Roy, D.B., Pywell, R. and Fuller, R.J. (2012), Population density but not stability can be predicted from species distribution models. Journal of Applied Ecology, 49: 581-590. https://doi.org/10.1111/j.1365-2664.2012.02138.x)

Title

The percentage of highly suitable habitat in the article title is a subjective classification. What you should analyze more is the trend of total suitable habitat under climate change scenarios. The threshold value for dividing suitable and unsuitable areas is not introduced in this paper. Please refer to the following research. (Susana Françaa, Henrique N. Cabral. 2019. Distribution models of estuarine fish species: The effect of sampling bias, species ecology and threshold selection on models' accuracy, Ecological Informatics, 51, 168-176; William T. Bean, Robert Stafford and Justin S. Brashares. 2012. The effects of small sample size and sample bias on threshold selection and accuracy assessment of species distribution modelsdoi: 10.1111/j.1600-0587.2011.06545.x).

2. Materials and methods

2.2. The collection of H. dorsalis distribution data

How much distributed data are obtained from different databases? How much distribution data was obtained from the field survey? If distribution data are obtained from published literature, references should be cited in this paper. I cannot find the distribution data of H. dorsalis in the GBIF database.

Consequently, a more accurate explanation of data selection would be required to make models more reliable and replicable and, if possible, distribution data should be made available. In Feng et al. (2019) (https://doi.org/10.1038/s41559-019-0972-5), there are useful information in order to improve this aspect.

2.3 Environmental variables

The authors used the environmental values extracted based on the distribution point data to do a correlation analysis between them in SPSS, which is obviously not reasonable. The correlation between bioclimatic factors refers to the correlation between layers. The correlation coefficients of the 19 bioclimatic factors were not provided by the authors, and in this paper the authors only filtered out 3 bioclimatic factors, which is obviously unreasonable. It is suggested that the authors use the ENMtools or ArcGIS to analyze the correlation of 19 bioclimatic factors.

2.4. Species Distribution Modeling

This study uses the maxent model with default parameter settings, which can lead to inaccurate results of the model. It is recommended to use ENMeval tool for parameter optimization. Kass, JM, Muscarella, R, Galante, PJ, et al. ENMeval 2.0: Redesigned for customizable and reproducible modeling of species’ niches and distributions. Methods Ecol Evol. 2021; 12: 1602– 1608. https://doi.org/10.1111/2041-210X.13628

3 results

The quality of the graphs in the article is far from publishable, e.g., Table 3 is an intercepted image, not a table. The resolution of Figure 7 is very poor.

4 Discussion

The authors discuss more about the reasons for the expansion of potentially suitable habitat of H. dorsalis under climate change. It would be more interesting to suggest that the authors explore and reveal the reasons and mechanisms behind the increase in population density of H. dorsalis by climate change and suggest preventive and control measures at the end of the paper.

Author Response

The manuscript analyzes the population density and high suitable area ratio of Hippotiscus dorsalis (Hemiptera: Pentatomidae) in China. The work has the potential of being an interesting study about the possible threat posed by H. dorsalis both in the present and in the future. However, several aspects of the paper need to be revised such as distribution data selection, methods explanation, etc. Moreover, the work lacks an ecological perspective and it would need a more in-depth analysis of bibliography regarding several topics. These are the main general issues:

This study contains two independent parts of research content, and the author does not connect the two parts. How to incorporate the population density change into the species distribution model?Combine the two parts is the main innovation of this study. (Tôrres, N.M., De Marco, P., Júnior, Santos, T., Silveira, L., de Almeida Jácomo, A.T. and Diniz-Filho, J.A.F. (2012), Can species distribution modelling provide estimates of population densities? A case study with jaguars in the Neotropics. Diversity and Distributions, 18: 615-627. https://doi.org/10.1111/j.1472-4642.2012.00892.x; Oliver, T.H., Gillings, S., Girardello, M., Rapacciuolo, G., Brereton, T.M., Siriwardena, G.M., Roy, D.B., Pywell, R. and Fuller, R.J. (2012), Population density but not stability can be predicted from species distribution models. Journal of Applied Ecology, 49: 581-590. https://doi.org/10.1111/j.1365-2664.2012.02138.x)

Revised. The relevant discussion of these two references was added to the study. Line 544. Although the present study is divided into two parts, they are closely linked by the topic of climate change. The design of the paper mainly considers the effects of climate change on the areas of general aspect (distribution) and key point (population density in a representative area).

Title

The percentage of highly suitable habitat in the article title is a subjective classification. What you should analyze more is the trend of total suitable habitat under climate change scenarios. The threshold value for dividing suitable and unsuitable areas is not introduced in this paper. Please refer to the following research. (Susana Françaa, Henrique N. Cabral. 2019. Distribution models of estuarine fish species: The effect of sampling bias, species ecology and threshold selection on models' accuracy, Ecological Informatics, 51, 168-176; William T. Bean, Robert Stafford and Justin S. Brashares. 2012. The effects of small sample size and sample bias on threshold selection and accuracy assessment of species distribution modelsdoi: 10.1111/j.1600-0587.2011.06545.x).

Revised. Line 213.

  1. Materials and methods

2.2. The collection of H. dorsalis distribution data

How much distributed data are obtained from different databases? How much distribution data was obtained from the field survey? If distribution data are obtained from published literature, references should be cited in this paper. I cannot find the distribution data of H. dorsalis in the GBIF database.

Consequently, a more accurate explanation of data selection would be required to make models more reliable and replicable and, if possible, distribution data should be made available. In Feng et al. (2019) (https://doi.org/10.1038/s41559-019-0972-5), there are useful information in order to improve this aspect.

Revised. Lines 135, 142. Occurrence records coord of Hippotiscus dorsalis in China was added in supplementary materials (Table S1).

2.3 Environmental variables

The authors used the environmental values extracted based on the distribution point data to do a correlation analysis between them in SPSS, which is obviously not reasonable. The correlation between bioclimatic factors refers to the correlation between layers. The correlation coefficients of the 19 bioclimatic factors were not provided by the authors, and in this paper the authors only filtered out 3 bioclimatic factors, which is obviously unreasonable. It is suggested that the authors use the ENMtools or ArcGIS to analyze the correlation of 19 bioclimatic factors.

ArcGIS and SPSS were utilized to calculate the correlation coefficients of 19 bioclimatic factors. However, the relevant description was revised. Line 166. “The absolute value of the correlation coefficient of greater than 0.9 was defined as being highly correlated” (line 171). In that part, 0.85, 0.9, and 0.95 were usually selected as the standards of high correlation. In the present research, a value greater than 0.9 was defined as being highly correlated; three bioclimatic factors met this criterion and were filtered out in this section. Even if 0.85 was selected as an appropriate value, three more bioclimatic factors (7, 12, and 17) would have been filtered out; the three bioclimatic factors did not contribute anything toward the prediction of the distribution of Hippotiscus dorsalis, and the result was not affected. The relevant references were also completed (line 156). Six bioclimatic factors (7, 8, 9 ,12, 16, and 17) were filtered out by the percentage contribution. Finally, 10 contributive environmental variables were selected. Less research articles put the correlation coefficients of the 19 bioclimatic factors in their articles, so it was not to show. Refer to the correlation coefficients of the 19 bioclimatic factors.xls.

2.4. Species Distribution Modeling

This study uses the maxent model with default parameter settings, which can lead to inaccurate results of the model. It is recommended to use ENMeval tool for parameter optimization. Kass, JM, Muscarella, R, Galante, PJ, et al. ENMeval 2.0: Redesigned for customizable and reproducible modeling of species’ niches and distributions. Methods Ecol Evol. 2021; 12: 1602– 1608. https://doi.org/10.1111/2041-210X.13628

Revised. The ENMeval tool was used to execute the parameter optimization observations. Figures 5-8 were reproduced according to the new results obtained; all related descriptions have been revised throughout the article.

3 results

The quality of the graphs in the article is far from publishable, e.g., Table 3 is an intercepted image, not a table. The resolution of Figure 7 is very poor.

 Revised. Table 3 was revised. Since an image has been added to the document, the resolution has reduced. High-resolution images will be uploaded as attachments.

4 Discussion

The authors discuss more about the reasons for the expansion of potentially suitable habitat of H. dorsalis under climate change. It would be more interesting to suggest that the authors explore and reveal the reasons and mechanisms behind the increase in population density of H. dorsalis by climate change and suggest preventive and control measures at the end of the paper.

Revised. Lines 516-580.

Author Response

Title: Climate change influences the population density and suitable area ratio of Hippotiscus dorsalis (Hemiptera: Pentatomidae) in China
This paper is aimed to find relationships between damage dynamics and climate factors, and predict current and future distribution of maso bamboo pest, (Hemiptera: Pentatomidae). This paper is largely revised, but there are still needed many revisions.
I do not find any difference between summary and conclusion. Hence, conclusion should be deleted if this journal permits.

Revised. The conclusion part was rewritten.
In text, many sentences with same meaning are repeated. Please check text carefully for nonredundant paper.

Revised.
In title, ratio is not needed.

Revised.
Line 12-13: the mean monthly temperature and maximum temperature -- the April mean
temperature and April maximum temperature

Revised.
Line 15-16: These findings provide a theoretical basis for the early forecasting and monitoring of
pest outbreaks. This sentence is not needed.

Revised.
Line 21: rule deleted.

Revised.
Line 37-38: bamboo shoots-eating (such as Otidognathus davidis), leaf-eating (such as Ceracris kiangsu), and stem-eating (such as Hippotiscus dorsalis) -- bamboo shoots-eating such as Otidognathus davidis, leaf-eating such as Ceracris kiangsu, and stem-eating such as Hippotiscus dorsalis

Revised.
Line 43: for this -- for this species

Revised.
Line 63-64: a distribution trend -- a distribution shift

Revised.
Line 74-75: Generalize – Generalized

Revised.
Line 107-108: Methods and instrument (model, cooperation) for measuring temperature is needed.
Only four temperatures are measured. How do you obtain maximum and minimum temperature?

Revised. The related description was added. U thermometer (Hebei Dongtai Instrument Co., China) was used to obtain maximum and minimum daily temperature.
Line 118: Error of double parentheses. )) -- )

This error is repeated in lines 141, 143, 171, and 188.

Revised.
Line 122-126, 133: If data from other sources are not used, the sentence in line 122-126 are not
needed.

Revised.
Line 163: Table 1 - Table 2?

Revised.
Table 2: Many Pearson significance should be deleted. Caption within table should be located out
of table.

Revised.
Line 179: What is logistic format?

Revised. A save format of predicting results. The related description was added.
Line 207-209: The number of moso bamboo plants surveyed was plants per year, and the survey
site was located in Anji County, Huzhou City, Zhejiang Province, China (Figure 1) This sentence is
repeated previously.

Revised.
Line 379-380: mean monthly temperature and the maximum -- April mean temperature and the
April maximum

Revised.
Line 396: all due to a sudden increase What this means? Please check English.

Revised.
Line 405: low altitude Altitude does not directly influence distribution of insects, but through
climatic factors related with altitude.

Revised.
Line 409: being Not needed.

Revised.
Line 431-434: There is a considerable amount of evidence that supports the notion that climate
change can have a significant impact on the geographical distribution of species [20,65,66]. The
strength of influence of climate change on species has a strong correlation with the location of the
species [67] I do not understand why these sentences are used.

Revised.
Line 452-454: In comparison to the current potential distribution of H. dorsalis, the total suitable area slightly decreased by 8.3% and 11.2% in 2050 and 2070, respectively, under RCP 2.6, attributed to the decline in the moderate suitable area. Should be deleted.

Revised.
Line 496: So Not needed.

Revised.
Line 509-510: mean monthly temperature and maximum temperature -- April mean temperature and April maximum temperature

Revised.

Reviewer 4 Report

The major problem with this paper is that too much unnecessary information is included in the paper.  An example is the number of correlations reported between various measurements of temperature.  The only correlations that are important are those between damage to bamboo and the number of H. dorsalis.  Therefore any temperature change that increases the population of H. dorsalis is important for understanding damage to bamboo.  When reporting the results that is the only result that should be emphasised.  More extensive editing is necessary to ensure that the paper is publishable.

Author Response

The major problem with this paper is that too much unnecessary information is included in the paper.  An example is the number of correlations reported between various measurements of temperature.  The only correlations that are important are those between damage to bamboo and the number of H. dorsalis. Therefor any temperature change that increases the population of H. dorsalis is important for understanding damage to bamboo.  When reporting the results that is the only result that should be emphasised.  More extensive editing is necessary to ensure that the paper is publishable.

The sample questions have been revised. Lines 251-258. The full text was revised.

Round 2

Reviewer 1 Report

The manuscript reports the dominant climatic factors affecting the damage and distribution of Hippotiscus dorsalis, and the potential distribution areas of H. dorsalis in China under current and future climate conditions. I add some opinions on the manuscript.

The superscript of authors is not needed because they work the same place.

Line 18: outbreak à outbreak of H. dorsalis

Line 26-28: Please rewrite this part.

Line 53: in late March –early April à from early March to early April.

Line 59-62: Logically, this sentence move to Line 71.

Line 73: delete “and”.

Line 106-107: the peak period of bamboo feeding by H. dorsalis was observed on April during the field observation.

Line 108: attacked bamboo rate à attacked rate of bamboo.

Line 123: National à national.

Line 137: Delete “of”.

Line 161: Please is the reference of ArcGIS and SPSS.

Line 172: for use à for using.

Line 194: was à is.

Line 219-220: The number of moso bamboo plants surveyed was 4400 plants per year.

Line 227: was the same as à was similar to.

Line 246: the highest and lowest points in 2005 and 2008, respectively.

Line 254: “were” à “was”.

Line 256: in April respectively à in April, respectively.

Line 260: in April respectively à in April, respectively.

Line 269: in April respectively à in April, respectively.

Line 269: Add “and” between “;” and “5)”.

Line 285: Figure 4 à Add period “Figure 4.”.

Line 292: “Bio6” à “bio6”.

Line 367: “2050 to 2070” à “2050 and 2070”.

Line 418-422: The authors described “The mean monthly temperature and maximum temperature were the main factors affecting population density and attacked bamboo rate”. Please give some discussion. Why “the mean monthly temperature and maximum temperature” became the main factors “affecting population density and attacked bamboo rate”.

Line 422-424: Would you give clear discussion? For example, what factors give the same results in Halyomorpha halys?

Line 450: “occurs” à “occurred”.

Line 453: “suitable climatic conditions” à Please give more detailed information.

Line 463: “Bio18” à “bio18”.

Line 466: “shows” à “showed”.

Line 501: “will shift” à “moved”

Line 505: “Halyomorpha halysàH. halys”.

Line 508: Hippotiscus dorsalis à H. dorsalis

Line 515: Hippotiscus dorsalis à H. dorsalis

Line 518: Hippotiscus dorsalis à H. dorsalis

Line 566-569: Please rewrite the sentence.

Line 580: to provide à and to provide.

Author Response

The superscript of authors is not needed because they work the same place.

Revised.

Line 18: outbreak à outbreak of H. dorsalis

Revised.

Line 26-28: Please rewrite this part.

Revised.

Line 53: in late March –early April à from early March to early April.

Revised.

Line 59-62: Logically, this sentence move to Line 71.

Revised.

Line 73: delete “and”.

Revised.

Line 106-107: the peak period of bamboo feeding by H. dorsalis was observed on April during the field observation.

Revised.

Line 108: attacked bamboo rate à attacked rate of bamboo.

Revised.

Line 123: National à national.

Revised.

Line 137: Delete “of”.

Revised.

Line 161: Please is the reference of ArcGIS and SPSS.

Revised.

Line 172: for use à for using.

Revised.

Line 194: was à is.

Revised.

Line 219-220: The number of moso bamboo plants surveyed was 4400 plants per year.

Revised.

Line 227: was the same as à was similar to.

Revised.

Line 246: the highest and lowest points in 2005 and 2008, respectively.

Revised.

Line 254: “were” à “was”.

Revised.

Line 256: in April respectively à in April, respectively.

Revised.

Line 260: in April respectively à in April, respectively.

Revised.

Line 269: in April respectively à in April, respectively.

Revised.

Line 269: Add “and” between “;” and “5)”.

Revised.

Line 285: Figure 4 à Add period “Figure 4.”.

Revised.

Line 292: “Bio6” à “bio6”.

Revised.

Line 367: “2050 to 2070” à “2050 and 2070”.

Revised.

Line 418-422: The authors described “The mean monthly temperature and maximum temperature were the main factors affecting population density and attacked bamboo rate”. Please give some discussion. Why “the mean monthly temperature and maximum temperature” became the main factors “affecting population density and attacked bamboo rate”.

Revised. Lines 424-428.

Line 422-424: Would you give clear discussion? For example, what factors give the same results in Halyomorpha halys?

Revised. Lines 424-430.

Line 450: “occurs” à “occurred”.

Revised.

Line 453: “suitable climatic conditions” à Please give more detailed information.

Lines 459-460.

Line 463: “Bio18” à “bio18”.

Revised.

Line 466: “shows” à “showed”.

Revised.

Line 501: “will shift” à “moved”

Revised.

Line 505: “Halyomorpha halys” à “H. halys”.

Revised.

Line 508: Hippotiscus dorsalis à H. dorsalis

Revised.

Line 515: Hippotiscus dorsalis à H. dorsalis

Revised.

Line 518: Hippotiscus dorsalis à H. dorsalis

Revised.

Line 566-569: Please rewrite the sentence.

Revised. These less irrelevant sentences were removed, and the limiting factors of the study were discussed. Lines 573-577.

Line 580: to provide à and to provide.

Revised.

Reviewer 2 Report

 This study contains two independent parts of research content, and the author does not connect the two parts. How to incorporate the population density change into the species distribution model?Combine the two parts is the main innovation of this study. (Tôrres, N.M., De Marco, P., Júnior, Santos, T., Silveira, L., de Almeida Jácomo, A.T. and Diniz-Filho, J.A.F. (2012), Can species distribution modelling provide estimates of population densities? A case study with jaguars in the Neotropics. Diversity and Distributions, 18: 615-627. https://doi.org/10.1111/j.1472-4642.2012.00892.x; Oliver, T.H., Gillings, S., Girardello, M., Rapacciuolo, G., Brereton, T.M., Siriwardena, G.M., Roy, D.B., Pywell, R. and Fuller, R.J. (2012), Population density but not stability can be predicted from species distribution models. Journal of Applied Ecology, 49: 581-590. https://doi.org/10.1111/j.1365-2664.2012.02138.x). I don't think your response can reasonably explain all t problem.

Author Response

References had been added to the original text in the previous revisions and related discussions were held. We still think that the design direction of this study and the reference are different. These two parts of the article are closely linked by the topic of climate change, which is significant for the prevention and control of H. dorsalis in China. Our study focuses on the effects of climate change on the areas of general aspect (distribution) and key point (population density in a representative area), instead of exploring the relationship between species distribution model and population density. Even if the revised suggestion was put forward at the beginning of our study and design, it was also hard to enforce. At the same time, it will consume a lot of manpower and resources to set up dozens of simultaneous investigations across the country just as the references, and the research is unaffordable. The areas with serious harm are mainly focused on high suitable area. Anji county is the representative of the high suitable area, and the survey results have implications for future prevention and control of H. dorsalis.

Reviewer 4 Report

Comments are on accompanying pdf.  May help to clarify some of the figures, especially such as the temperature versus year as the graphs could be combine to make the patterns clearer with the y-axis having equivalent ˚C.

Author Response

The version was revised according to these relevant comments, and a few further explanations are as follows:

(1) Figure 2 and 3 were redone according to suggestions and the clear pictures were uploaded separately.

(2) The phrases in the following sentences were checked and correctly stated.

U thermometer (Line 101) is the name of the thermometer used; the total radiative forcing of 2100 (Line 137); the high suitable area (Line 444); 19 environment variables (Line 481).

(3) Figure 8: Are these figures significantly different as no variation is shown and the values are not very different.

Answer: The mapping and analysis refer to relevant research [1,2,3]. The relevant research analyzed the trend of change, and the significant difference was not analyzed.

  1. Ning, H.; Tang, M.; Chen, H. Impact of climate change on potential distribution of Chinese White Pine Beetle Dendroctonus armandi in China. Forests 2021, 12, 544. https://doi.org/10.3390/f12050544
  2. Ning, H.; Tang, M.; Chen, H. Mapping invasion potential of the pest from Central Asia, Trypophloeus klimeschi(Coleoptera: Curculionidae: Scolytinae), in the shelter forests of Northwest China. Insects 202112, 242. https://doi.org/10.3390/insects12030242
  3. Li, X.; Ge, X.Z.; Chen, L.H.; Zhang, L.J.; Wang, T.;  ZongX. Climate change impacts on the potential distribution of Eogystia hippophaecolus in China. Pest Manag. Sci. 2019, 75, 215-223. https://doi.org/10.1002/ps.5092

Round 3

Reviewer 2 Report

The logic of this article is not smooth, for example, the content of lines 38-41 does not match the main idea of the article, and lines 52-55 are not related to the content of the first sentence of the second paragraph. What exactly does "other conditions" in line 67 refer to?

The diagrams in this paper are far from the publication standard, for example, the font is not of uniform type, the legend is not uniform, the scale is not standardized (Figs. 4, 6, 7), and the raster layers and country boundaries do not match (Fig. 4).

Translated with www.DeepL.com/Translator (free version)

Translated with www.DeepL.com/Translator (free version)

Author Response

The logic of this article is not smooth, for example, the content of lines 38-41 does not match the main idea of the article,

Revised.

and lines 52-55 are not related to the content of the first sentence of the second paragraph.

Revised.

What exactly does "other conditions" in line 67 refer to?

Revised. Factors other than climate, for example food, natural enemy and so on.

The diagrams in this paper are far from the publication standard, for example, the font is not of uniform type, the legend is not uniform, the scale is not standardized (Figs. 4, 6, 7), and the raster layers and country boundaries do not match (Fig. 4).

Revised. Figure 4 was renewed.